# Pullulan/Poly(Vinyl Alcohol) Composite Hydrogels for Adipose Tissue Engineering

**DOI:** 10.3390/ma12193220

**Published:** 2019-10-01

**Authors:** Iuliana Samoila, Sorina Dinescu, Gratiela Gradisteanu Pircalabioru, Luminita Marutescu, Gheorghe Fundueanu, Magdalena Aflori, Marieta Constantin

**Affiliations:** 1Department of Biochemistry and Molecular Biology, University of Bucharest, 91-95 Splaiul Independentei, 050095 Bucharest, Romania; iuliana.samoila@bio.unibuc.ro (I.S.); sorina.dinescu@bio.unibuc.ro (S.D.); 2Research Institute of University of Bucharest, University of Bucharest, 050107 Bucharest, Romania; lumi.marutescu@gmail.com; 3Sanimed International IMPEX SRL, Sos. București Măgurele 70F, 051434 Bucharest, Romania; 4Department of Natural Polymers, Bioactive and Biocompatible Materials, Institute of Macromolecular Chemistry, 700487 Iassy, Romania; gfundeanu@icmpp.ro (G.F.); maflori@icmpp.ro (M.A.)

**Keywords:** hydrogels, poly(vinyl alcohol) (PVA), pullulan, scaffolds, adipose tissue engineering

## Abstract

Composite hydrogels based on pullulan (HP) and poly(vinyl alcohol) (PVA) were both prepared by simple chemical crosslinking with sodium trimethaphosphate (STMP) or by dual crosslinking (simultaneously chemical crosslinking with STMP and physical crosslinking by freeze-thaw technique). The resulting hydrogels and cryogels were designed for tissue engineering applications. PVA, with two different molecular weights (47,000 and 125,000 g/mol; PVA_47_ and PVA_125_, respectively), as well as different P/PVA weight ratios were tested. The physico-chemical characterization of the hydrogels was performed by FTIR spectroscopy and scanning electron microscopy (SEM). The swelling kinetics, dissolution behavior, and degradation profiles in simulated physiological conditions (phosphate buffer at pH 7.4) were investigated. Pullulan concentration and the crosslinking method had significant effects on the pore size, swelling ratio, and degradation profiles. Cryogels exhibit lower swelling capacities than the conventional hydrogels but have better stability against hydrolitic degradation. Biocompatibility of the hydrogels was also investigated by both MTT (3-(4,5-dimethylthiazol-2-yl)-2,5-diphenyltetrazolium bromide) and LDH (lactaten dehydrogenase) assay. The MTT and LDH assays proved that dual crosslinked HP/PVA_125_ (75:25, *w*/*w*) scaffolds are more biocompatible and promote to a greater extent the adhesion and proliferation of L929 murine fibroblast cells than chemically crosslinked HP/PVA_47_ (50/50, *w*/*w*) scaffolds. Moreover, the HP/PVA_125_ cryogel had the best ability for the adipogenic differentiation of cells. The overall results demonstrated that the HP/PVA composite hydrogels or cryogels are suitable biomaterials for tissue engineering applications.

## 1. Introduction

Modern healthcare constantly has to deal with an increased number of patients with high-grade burns or patients who need tissue reconstruction after tissue loss or tumor removal [1]. The first attempts at replacing damaged tissue were made using techniques based on autologous or allogenic transplants or with soft tissue fillers [2]. These techniques are not always the best answer for the regeneration of severe adipose tissue defects because of the adverse effects they may cause, such as inflammation or structure deformation [3]. 

Therefore, adipose tissue engineering (ATE) has emerged as a method to overcome all these disadvantages. For ATE to be successful, there are two characteristics that need to be combined; a cell source able to differentiate towards adipogenic lineage and a tridimensional *scaffold* with an interconnected pore architecture, so that oxygen and nutrients can reach the cells [3,4]. Materials used in ATE have to be soft and easy to operate with, in order to minimize patient discomfort [5]. Furthermore, because of the highly vascularized adipose tissue, the biomaterial should support proper vascularization to ensure the transportation of vital elements for adipocytes; growth factors, cytokines, and hormones.

Hydrogel scaffolds are used to provide massive and mechanical structures for a tissue construct, whether the cells are suspended or adherent to the three-dimensional hydrogel. Hydrogels can be obtained from both synthetic [6] and natural [7] polymers by various methods such as physical, chemical gelation, or self-assembly. In the last years, hydrogels prepared from natural polymers, especially polysaccharides [8], have been widely used because of their hydrophilicity, biocompatibility, and low toxicity. Among them, pullulan (P) has been increasingly studied in vascular and bone tissue engineering. Thus, porous scaffolds based on dextran and pullulan support the viability, proliferation, differentiation, and function of human endothelial cells isolated from heart blood [9] and composite hydroxyapatite/pullulan/dextran hydrogels induce cell differentiation in vitro and formation of mineralized bone tissue in vivo [10]. Collagen-pullulan composite scaffolds viably sustain in vitro human fibroblasts and murine mesenchymal stem cells and endothelial cells [11].

On the other hand, poly(vinyl alcohol) (PVA)-based hydrogels are promising candidates for tissue engineering applications. PVA is a biosynthetic polymer, biocompatible and non-toxic, with a great ability to form hydrogels either through chemical or physical crosslinking. While chemical crosslinking with radiation or aldehydes provides greater control over the final properties of the hydrogel, physically crosslinked hydrogels, or blends with other biocompatible polymers (dextran [12,13], starch [14], chitosan [15,16], alginates [17,18,19], gelatin [20,21], poly(vinyl pirrolidone) [22] (are more suitable candidates for biomedical applications. Scaffolds based on PVA were proposed for long-term culture of hepatocytes and mesenchymal stem cells [23,24,25]. 

Human adipose-derived stem cells (hASCs) represent a population of mesenchymal stem cells, which can be easily isolated from liposuction aspirate and cultured. They are often used in tissue engineering applications because of their tendency to differentiate into pre-adipocytes [3,4]. 

Adipocytes’ cytoplasm contains a large number of lipid droplets, which have a major importance in cellular lipid homeostasis by storing triacylglycerols [26]. Lipid droplets have an external phospholipid monolayer of lipid droplet proteins, including perilipin. Perlipin is a key component in lipid storage and is the most common protein found on the surface of lipid droplets [27] (Sawada et al., 2010). Perilipin is of great interest when evaluating adipogenesis, because its expression is higher when preadipocytes differentiate into adipocytes [28].

As seen in the literature, both P [29,30] and PVA [31] suffer chemical crosslinking with trisodium trimethaphosphate (TMP), a non-toxic cyclic triphosphate crosslinking agent already used in hydrogel synthesizing for pharmaceutical uses. Since the chemical crosslinked PVA presents poor elasticity and resistance [32] we propose a combining method of chemical agent and freeze-thawing cycles for obtaining composite HP/PVA hydrogels with a high hydration degree and good mechanical properties. The hydrogels were characterized in terms of swelling ratio and dissolution behavior. Finally, biocompatibility and adipogenic differentiation ability of the novel hydrogels were assessed.

## 2. Materials and Methods

### 2.1. Materials

Pullulan (P) (Mw = 200,000 g/mol) was purchased from Hayashibara Lab. Ltd. (Okoyama, Japan). Poly(vinyl alcohol) (PVA_125_) (average Mw ~ 125,000 g/mol, degree of hydrolysis 98.0–98.8 mol%), (PVA_47_) (average Mw ~ 47,000 g/mol, degree of hydrolysis 98.0–98.8 mol%), trisodium trimethaphosphate p.a. (STMP), and sodium hydroxide p.a. (NaOH) were purchased from Sigma–Aldrich Chemie GmbH, Steinheim, Germany. 

### 2.2. Methods

#### 2.2.1. Preparation of HP/PVA Crosslinked Composite Hydrogels

Composite HP/PVA hydrogels were obtained by crosslinking reaction of the two components with STMP in alkaline conditions. Firstly, 10% (*w*/*v*) PVA_47_ or 5% (*w*/*v*) PVA_125_ aqueous solutions were prepared by heating at 90 °C for 1 hour under magnetic stirring. Then, the corresponding amount of pullulan was added so as to obtain the gravimetric ratio between the two polymers shown in Table 1. The solution was cooled to room temperature, then 1 mL of 10M NaOH and 5 mL of a 10% (*w*/*v*) STMP aqueous solution were added. The solution was kept under vigorous stirring for several minutes after which it was poured into 6 cm diameter petri dishes. Finally, the solutions were either subjected to the freezing-thawing process with 3 cycles of 20 hours at 20 °C and 6 hours at room temperature or were maintained at room temperature (~23 ± 1 °C) for the same time (Table 1). At the end of the cycles, the disc hydrogels were extensively washed with distilled water then recovered by lyophilization (57 °C, 5.5 × 10^−4^ mbar) using a lyophilizer ALPHA 1-2 LD, Christ, Germany. The sample codes are HP/PVAx-R or HP/PVAx-F, where P means pullulan, PVA is poly(vinyl alcohol), x = molecular mass of PVA, R and F refer to the crosslinking procedure used (room temperature (R) or freeze-thawing (F)). 

#### 2.2.2. FTIR Spectroscopy

The spectra were recorded using a Bruker Vertex 7 FTIR spectrophotometer. The samples were analyzed in lyophilized state on a KRS-5 support, within the frequency range of 4000–600 cm^−1^. Data processing was done using the OPUS 6.5 software (Bruker Optik GmbH, Ettlingen, Germany).

#### 2.2.3. Morphology 

Morphological characterization was performed by scanning electron microscopy (SEM,) using an environmental scanning electron microscope (ESEM, FEI, Eindhoven, The Netherlands) type Quanta 200, operating with secondary electrons in low vacuum at 20 kV.

#### 2.2.4. Swelling and Dissolution Behavior

The swelling degree of the hydrogels was measured gravimetrically in phosphate buffer pH 7.4 (PBS). The process involves immersing dry weighed hydrogels (*W*_d_) in the swelling medium and weighing them at different time intervals (*W*_s_) after they have been removed from the test medium and buffered lightly onto the surface with filter paper. The experiment was performed in triplicate and average values were reported. Swelling ratio (SR) was calculated according to Equation (1):(1)$$SR=\frac{\left(W_{s}\;-\;W_{d}\right)}{W_{d}}.$$

The stability of HP/PVA hydrogels upon swelling in water was examined. Each sample of raw hydrogel (d × h = 1⋅× 5 cm) was immersed in water for 3 days at room temperature (23 ± 1 °C). After that the samples were removed from the water and then the excess water and degraded polymer were removed from the surface of the hydrogels through blotting. Samples were weighed before and after immersion, and the dissolution percent (D %) was calculated according to Equation (2) [33] (Abdel–Mohsen et al., 2011): (2)$$D\;\left(\%\right)=\frac{W_{2}}{W_{1}}\times 100$$where *W*_2_ and *W*_1_ are the weight of the sample after and before extraction, respectively.

#### 2.2.5. In Vitro Degradation Studies

Degradation of the hydrogel was evaluated by incubating dried samples (2 × 1 cm) in PBS at 37 °C. The PBS solution was changed weekly. At pre-selected time points, the remaining hydrogels were removed from the buffer solution, washed with distilled water, lyophilized, and dried in a vacuum drier at 60 °C to constant weight. The percent of weight loss (WL) of each sample was determined according to Equation (3):(3)$$WL\;\left(\%\right)=\frac{\left(W_{0}-W_{t}\right)}{W_{0}}\times 100$$ where *W*_0_ and *W*_t_ are the sample weights before and after degradation, respectively. All data were averaged from three measurements.

#### 2.2.6. Biocompatibility Studies

All composite samples were sterilized by exposure to UV light for 24h, then the materials were put into contact with complete medium and cut into small pieces of 1 cm diameter. After 2 and 6 days of culture in standard conditions, L929 fibroblast cell viability and proliferation were assessed quantitatively by MTT test and qualitatively by Live/Dead staining and observation by confocal microscope. The three-dimensional cultures were incubated with 1 mg/mL MTT solution (Sigma–Aldrich Co, Steinheim, Germany) in culture media with no fetal bovine serum (FBS). After 4 h of incubation, formazan crystals were solubilized in isopropanol, resulting in a violet solution that was quantified by spectrophotometry at 550 nm using FlexStation3 (Molecular Devices, USA). Scaffold’s cytotoxic effect on the cells was measured using an In Vitro Toxicology Assay Kit, Lactic Dehydrogenase based (Tox7 kit, Sigma–Aldrich) following the manufacturer’s instructions. The final solution was assessed by spectrophotometric measurement at 490 nm. Staining of live and dead cells in the three-dimensional cultures was performed following the manufacturer’s protocol using a Live/Dead kit (ThermoFisher Scientific, Foster City, CA, USA). Composites were examined by confocal microscopy (Carl Zeiss LSM 710, Jena, Germany), and images were processed using Zeiss Zen 2010 software (2010 version, Carl Zeiss, Jena, Germany). 

#### 2.2.7. Adipogenic Differentiation Assessment

(1) Adipogenic marker perilipin gene expression analysis

Total RNA was extracted from the bioconstructs with TRIzol Reagent, and was then tested for purity and integrity by Agilent Bioanalyzer 2100. Reverstranscription was performed using a High Capacity cDNA Reverse Transcription Kit (ThermoFisher Scientific, Foster City, CA, USA) and final quantitative Real Time PCR(qPCR) was performed on the ViiA7 system using SYBR Green Master Mix (ThermoFisher Scientific, Foster City, CA, USA), following the manufacturer’s instructions. 

(2) Adipogenic marker perilipin protein expression 

Cells were fixed using 4% paraformaldehyde solution (Sigma–Aldrich Co, Steinheim, Germany) for 2 h and then permeabilized for 2 h with 2% BSA/0.1% Triton X-100 solution (Sigma–Aldrich Co., Steinheim, Germany). Bioconstructs were exposed overnight at 4 °C to rabbit polyclonal anti-perilipin antibody solution (1:100, Santa Cruz Biotechnology). The next day, secondary antibody goat anti-rabbit, coupled with AF546 (1:400, ThermoFisher Scientific, Foster City, CA, USA), was incubated for 2 h at room temperature, in darkness. Hoechst 33342 (ThermoFisher Scientific, Foster City, CA, USA) was used for 30 minutes to stain cell nuclei. Bioconstructs were observed using confocal microscopy (Carl Zeiss LSM 710, Jena, Germany) and images were analyzed with Zeiss Zen 2010 software. Quantification of fluorescent staining was performed by Image J software (version 1.52q, National Institutes of Health, Bethesda, Maryland, U.S.) and data is presented in percentage of area covered by red-labeled pixels.

#### 2.2.8. Statistical Analysis

All experiments were performed in triplicate. Statistical analysis was performed using Graph Pad Prism software, one –way ANOVA method, and Bonferroni correction. Statistical significance was considered for *p* < 0.05.

## 3. Results and Discussion

### 3.1. Preparation and Characterization of HP/PVA Composite Hydrogels

PVA has already been combined with P in applications in cosmetics and drug delivery [34,35]. Both polymers contain hydroxyl groups that can be easily chemically crosslinked with STMP. Moreover, supplementary crosslinking by cryogelation gives scaffolds improved hydrophilicity and mechanical strength. In alkaline conditions, the crosslinking reaction with STMP leads to the formation of biodegradable phosphoester bridges and phosphate pendant groups, negatively charged, favoring a high swelling in water [36,37]. The chemical crosslinking of the two polymers with STMP led to a transparent hydrogel (Figure 1A) whereas additional cryogelation generated opaque hydrogels (Figure 1B); a distinguishing sign for the physical crosslinking of PVA. The composition of HP/PVA hydrogels prepared by different crosslinking methods is presented in Table 1.

#### 3.1.1. Morphology

SEM images of the internal structure of hydrogels obtained by chemical crosslinking with STMP and by supplementary crosslinking by cryogelation are presented in Figure 2. All types of hydrogels display a porous structure with interconnected pores. The size of the pores decreases from 30–55 μm for HP/PVA_47_-R to 15–30 μm for HP/PVA_47_-F and HP/PVA_125_-F. In fact, the freezing–thawing procedure induced the crystallization of the polymeric chains of PVA and resulted in a denser network structure, which act as a physical crosslinker. However, through the freeze-thaw technique, there were non-homogeneous networks with areas where PVA can be assumed to be concentrated due to its physical crosslinking (Figure 2, HP/PVA_47_-F and HP/PVA_125_-F).

#### 3.1.2. Chemical Structure

Crosslinking with STMP, and therefore the presence of phosphate groups in the HP/PVA hydrogels, was revealed by Fourier transform infrared spectrometry (FTIR). In Figure 3, the infrared spectra of native PVA and HP/PVA hydrogels prepared by the two techniques are presented and compared. The FTIR spectrum of the native PVA exhibits the typical stretch band of the inter- and intra-molecular OH groups at 3430 cm^−1^, the stretching bands corresponding to the symmetrical and asymmetric CH_2_ bonds at 2928 and 2855 cm^−1^, respectively, and a band at approximately 1720 cm^−1^, which can be attributed to the presence of the acetoxy group [38]. The IR spectrum of HP/PVA hydrogels shows the presence of the absorption bands characteristic of the two polymers but also the appearance of new bands, confirming crosslinking with STMP. Thus, the inter- and intra-molecular bonds in which the OH groups are involved are evidenced by the presence of the band at 3300 cm^−1^, and the new significant bands in the 1307–1206 cm^−1^ region are representative of the presence of the pyrophosphate systems. The adsorption bands corresponding to the phosphate groups appear in the 900–1140 cm^−1^ region but overlap with those specific to the pullulan. However, one shoulder is present at 1139 cm^−1^. The disappearance of the 1720 cm^−1^ band, which is specific to the acetoxy group, is due to the alkaline environment in which the crosslinking reaction was performed. P-O-C bonding in the region between 970 and 1050 cm^−1^ can also be verified, confirming the presence of crosslinks between PVA and STMP [39].

The diminution of the peaks from 2935 and 2855 cm^−1^, corresponding to the asymmetrical and symmetrical stretching of the CH_2_ of PVA, suggests crystallite formation in which the mobility of these groups are retained [40].

### 3.2. Swelling and Dissolution Behavior

The swelling degree is a process influenced both by the degree of crosslinking and the electrostatic rejection between macromolecular chains. 

Since hydrogels have been prepared in the present study both by chemical crosslinking at room temperature and by chemical and physical crosslinking by freezing/thawing, the swelling process is expected to be influenced mainly by the crosslinking procedure and the crosslinker itself. The presence of the crosslinker (STMP) in a higher amount in the polymeric network has two opposing effects: i) It increases the number of intermolecular bridges, reducing the swelling degree and ii) It increases the number of anionic charged groups (phosphate) in the hydrogel, generating electrostatic rejections between neighboring macromolecular chains, thus increasing the swelling.

In the case of cryogels, water can enter through the interconnected pores of the network by convection. In contrast to the cryogels, the swelling kinetics of the hydrogels are controlled by the diffusion of solvent molecules through the gel network, which is a slow process [41].

Therefore, it can be observed that HP/PVA cryogels swell less than the corresponding hydrogels. This is expected since the cryogels are formed below the freezing point of water, and by freeze/thaw cycles, a denser crosslinked network and harder hydrogels are obtained [41]. 

Consequently, the swelling kinetics of the two types of hydrogels are different (Figure 4A). The HP/PVA_47_-R hydrogels gradually take up to 6 hours to swell due to their high porosity and uniform structure of the network. On the contrary, the HP/PVA-F cryogels swell rapidly, reaching equilibrium in approximately 60 min. It can be seen that the HP/PVA_125_-F cryogels swell more than the HP/PVA_47_-F one. This behavior is as expected since, for the HP/PVA_125_ preparation, we used PVA with higher molecular weight and in a smaller final concentration compared with the HP/PVA_47_-F cryogels (see Table 1). On the one hand, the lower concentration of PVA in the HP/PVA_125_-F cryogel led to a structure that was less crosslinked, with slightly larger porosity (Figure 2), which increased the equilibrium swelling [42]. Hassan and Peppas (2000) found that, for hydrogels obtained by using higher molecular weight PVA, higher volume swelling ratios are obtained due to an additional crystallization during swelling and increased mobility because of less physical crosslinking [43].

Upon placement in water for 3 days, all HP/PVA samples prepared lost between 4 and 10 percent of their structure. From the data presented in Figure 4B, it can be assumed that a more stable structure is obtained by combining the two crosslinking methods, since a smaller fraction of P and PVA chains that were not incorporated into the overall structure of P/PVA_47_-F hydrogel (4%) dissolved into water compared with P/PVA_47_-R hydrogel (8.2%). A lower degree of PVA and P dissolution was observed with samples prepared with lower molecular weight PVA and by using higher PVA concentrations (D% for HP/PVA_47_-F < D% for HP/PVA_125_-F). Therefore, it is evident from these results that there is an increased degree of physical crosslinking associated with an increased initial PVA concentration. However, the apparent difference between the dissolution percentages of the three types of hydrogels is not significant if SD values are considered.

### 3.3. In Vitro Degradation Studies

Ideally, hydrogels as tissue engineering scaffolds should be degradable at a rate proportional to the formation of new tissue [42]. The degradation profiles of the HP/PVA hydrogels after 28 days (expressed as a % weight loss) are illustrated in Figure 5. As apparent from the figure, controlled mass loss rates (less than 8% weight loss) were displayed in the HP/PVA composite cryogels (4.83 ± 0.5% for HP/PVA_47_-F and 7.19 ± 0.8% for HP/PVA_125_-F, respectively). When we compare the composite cryogels after 28 days, HP/PVA_125_-F showed a statistically significant increase (p < 0.02). After the same period, the degree of degradation of HP/PVA_47_-R hydrogels was found to be 12.4 ± 1.5 %, higher that for the corresponding cryogels; this hydrogel displayed a statistically significant augmentation of weight loss compared with the HP/PVA_47_-F (*p* < 0.001) and HP/PVA_125_-F (*p* < 0.007) cryogels. These observations showed that the stability of the hydrogels increased by combining the two crosslinking methods. Obviously, as the pullulan concentration increased, the degradation rate increased, since only the polysaccharide is susceptible to degradation.

### 3.4. Biocompatibility

After 2 days of culture in standard conditions, MTT assay results indicated a good biocompatibility on all tested materials, with a slightly increased viability for HP/PVA_125_-F when compared to HP/PVA_47_-F and HP/PVA_47_-R materials (Figure 6A). HP/PVA_47_-F and HP/PVA_47_-R had similar viability values with the control. After being in culture for 6 days, the composites started showing different levels of proliferation. The control system showed a statistically significant higher proliferation (*p* < 0.01) at 6 days in comparison to the level registered at 2 days. The same results were observed for cells cultured on HP/PVA_47_-F. HP/PVA_125_-F, which showed a better predisposition for increased viability at first, at 6 days displayed a statistically significant elevation of proliferation (*p* < 0.001). When comparing the proliferation at 6 days on all three tested composites, HP/PVA_125_-F showed the highest statistically significant increase (*p* < 0.01). Furthermore, a slight tendency of increased viability, with no statistical significance, was noticed for cells cultured on HP/PVA_47_-F material in comparison with the HP/PVA_47_-R scaffold. By supporting cell proliferation the best, HP/PVA_125_-F indicated that PVA with P could promote and stimulate cell viability and proliferation.

Composites’ cytotoxicity was assessed by LDH assay in order to quantitate the level of LDH enzyme released in the cell culture media. Results showed that after 2 days of culture, the scaffolds exhibited similar cytotoxicity compared to the control (Figure 6B). At 6 days, the cytotoxicity of the materials slightly increased but remained at approximately the same level as the control. No statistically significant difference was found between the LDH levels released by the tested composites, indicating that the combination of PVA with P does not exert an important cytotoxic effect on the cellular component.

Confocal microscopy was used to evaluate Live/Dead staining, which was performed in order to visualize cell dispersion inside the materials and the proportion of live (green) and dead (red) cells. The results indicated a consistency with the MTT and LDH assays, because, overall, the number of viable cells was much higher than the number of dead cells. Additionally, visualization by confocal microscopy allowed the observation of cell morphology, which was characteristic for the three-dimensional culture systems (Figure 6C). After 2 days of culture in standard conditions, a high proportion of live cells was observed on all scaffolds. Furthermore, cells were evenly distributed on the control, HP/PVA_47_-F, HP/PVA_125_-F, and HP/PVA_47_-R. After 6 days of being in contact with the composites, the highest number of viable cells (Figure 6) was established on HP/PVA_125_-F when compared to the control and to the HP/PVA_47_-F and HP/PVA_47_-R materials. Even though, at 2 days, HP/PVA_47_-R had a similar number of live cells to the rest of the composites, after 6 days of culture the percentage of dead cells increased compared to the other composites, indicating that its composition has negative effects on cell behavior.

### 3.5. Adipogenic Differentiation Assessment

#### 3.5.1. Adipogenic Marker Perilipin Gene Expression Analysis

Evaluation of perilipin was assessed in order to investigate the adipogenic differentiation of cells, since perilipin is a late differentiation marker. Consequently, its level was expected to be low seven days after adipogenesis induction. After 7 days of differentiation, perilipin gene expression for cells cultivated on HP/PVA_47_-F and HP/PVA_125_-F materials was similar, but for cells cultivated on HP/PVA_47_-R it was slightly lower, with no statistical significance (Figure 7A). After cells were kept in culture with a differentiation medium for 21 days and then compared to the results obtained at 7 days, perilipin gene expression was found to be significantly higher for cells cultivated on all tested materials (overall, approximately 3 times higher). In the case of cells differentiated in contact with HP/PVA_125_-F for 21 days, a statistically significant increased expression (*p* < 0.001) was found when compared to perilipin levels at 7 days and to the other composites. The same results were observed for cells differentiated on the control system (*p* < 0.001), with perilipin being highly expressed after 21 days of induced adipogenesis. Perilipin expression was the highest on the control system because the cells seeded on the tissue culture plastic system undergo adipogenic differentiation at a faster rate than the ones cultivated on materials. Cells cultivated in contact with HP/PVA_47_-F and HP/PVA_47_-R for 21 days, expressed perilipin in a statistically significant manner (*p* < 0.01). 

#### 3.5.2. Adipogenic Marker Perilipin Protein Expression

Similar to the gene expression, protein expression of adipogenic marker perilipin was evaluated at 7 and 21 days of adipogenic differentiation. Perilipin is a late differentiation marker, as its expression is higher when pre-adipocytes are fully differentiated into adipocytes. Confocal microscopy indicated that perilipin was present in cells cultured in contact with all tested materials after 7 days (Figure 7B). Perilipin expression was detected in cells cultured in contact with HP/PVA_47_-F after 7 days, but it was highly expressed after 21 days of differentiation. The same results were obtained for HP/PVA_47_-R, where perilipin was highly expressed after cells were kept in contact with the material for 21 days. These two scaffolds, HP/PVA_47_-F and HP/PVA_47_-R, presented a comparable expression of perilipin, whereas the most significant expression was visualized in cells cultured in contact with HP/PVA_125_-F after 21 days. When performing the quantification of the area and the intensity of red-labeled perilipin by Image J, a clear higher expression of perilipin was obtained for HP/PVA125-F than on the other compositions, confirming that this scaffold supported adipogenesis with the highest efficiency.

## 4. Conclusions

Porous composite pullulan/PVA hydrogels and cryogels were successfully prepared using a combining chemical and physical crosslinking at room temperature method or by a freeze–thawing method. The materials described in this study have shown good swelling capacity, as well as excellent stability against dissolution in water or degradation in PBS pH 7.4 at 37 °C. The cryogels also demonstrated very good biocompatibility, which makes them promising materials for tissue engineering because of their adipogenic differentiation ability.

It must be underlined that the excellent biocompatibility of PVA and P, combined with a mild procedure and the avoidance of any toxic crosslinking molecules, recommend the use of these hydrogels in biomedical applications.

## Figures and Tables

**Figure 1 materials-12-03220-f001:**
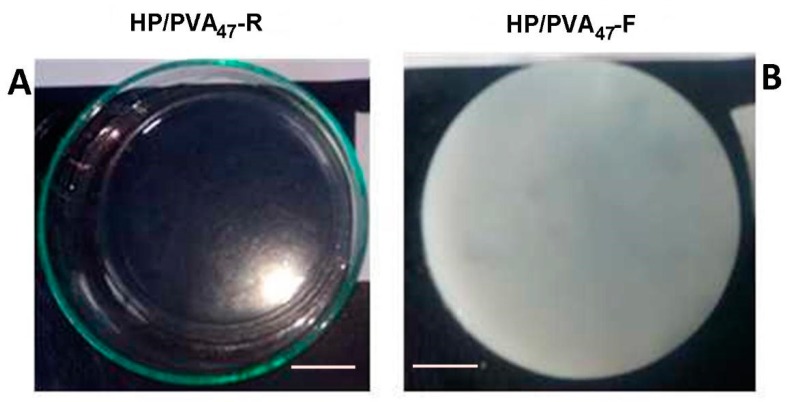
Optical images of HP/PVA_47_ composite hydrogels obtained by chemical crosslinking at room temperature (panel **A**) and by combined procedure (chemical and freeze-thawing cycles) (panel **B**). Bar corresponds to 1 cm.

**Figure 2 materials-12-03220-f002:**
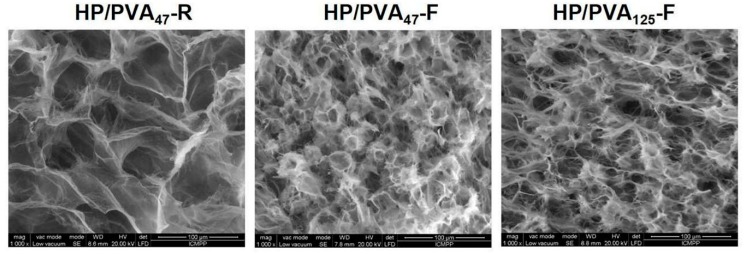
SEM microphotographs of the internal structure of composite hydrogels obtained by chemical crosslinking with STMP at room temperature (HP/PVA_47_-R) and by combined procedure (chemical and freeze-thawing cycles) (HP/PVA_47_-F and HP/PVA_125_-F).

**Figure 3 materials-12-03220-f003:**
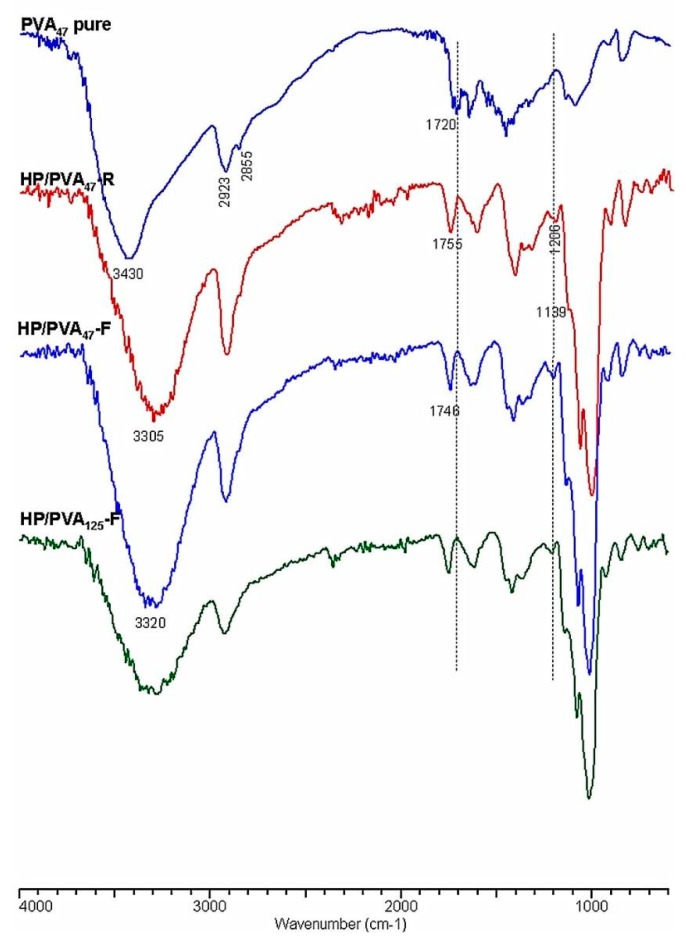
FTIR spectra of PVA and HP/PVA hydrogels.

**Figure 4 materials-12-03220-f004:**
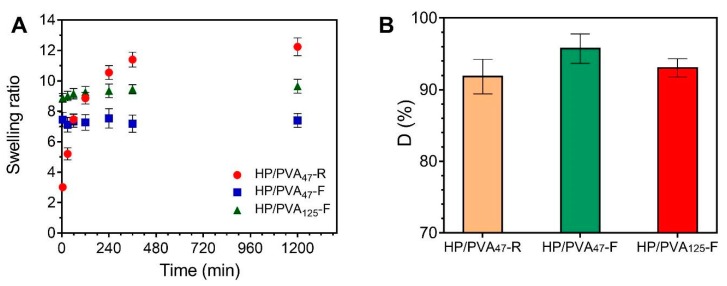
Swelling kinetics of HP/PVA composite hydrogels in simulated physiological conditions (PBS at 37 °C) (**A**) and dissolution percent in water at 23 °C of composite hydrogels (**B**).

**Figure 5 materials-12-03220-f005:**
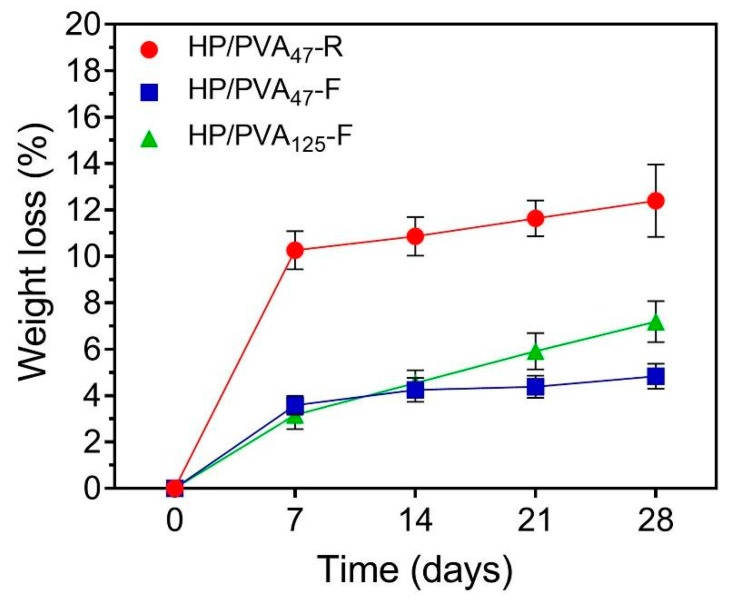
Weight loss profiles of HP/PVA composite hydrogels in PBS at 37 °C.

**Figure 6 materials-12-03220-f006:**
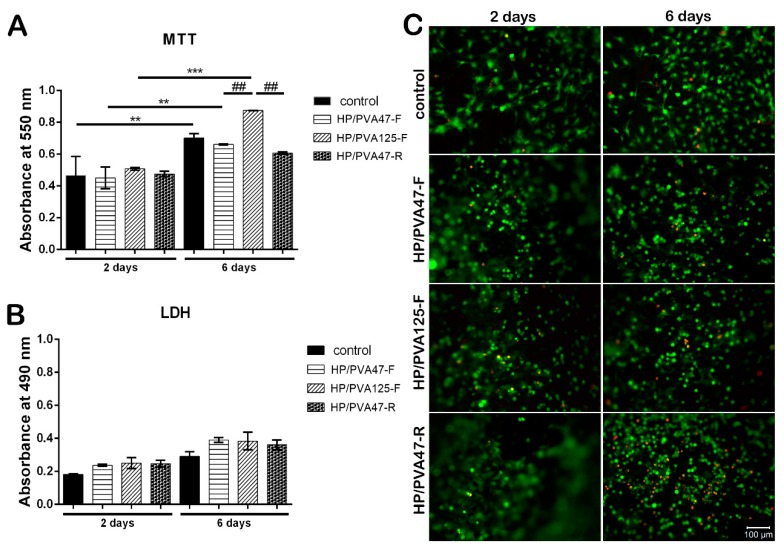
Quantitative evaluation of cell viability and proliferation after 2 and 6 days of culture, using MTT assay (**A**); Quantitative evaluation of the material’s cytotoxicity after 2 and 6 days of culture, using LDH assay (**B**); Confocal microscopy displaying live (green) and dead (red) cells after 2 and 6 days of culture (**C**); scale bar 100 µm.

**Figure 7 materials-12-03220-f007:**
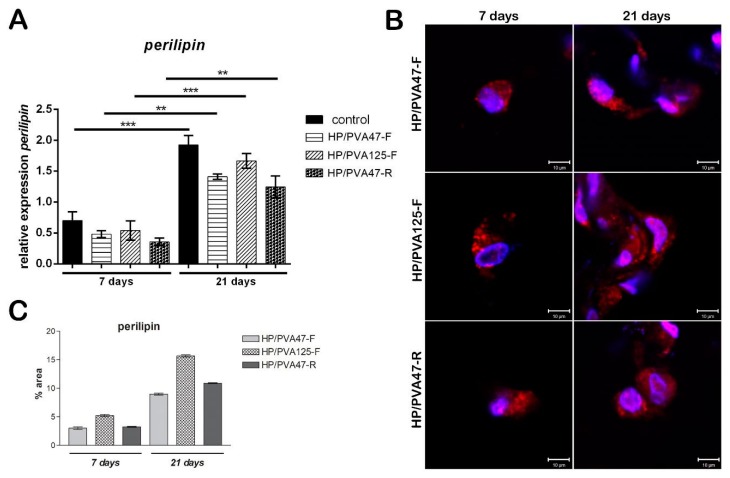
Evaluation of perilipin gene expression after 7 and 21 days of adipogenic differentiation (**A**); Evaluation of perilipin protein expression after 7 and 21 days of adipogenic differentiation by confocal microscopy (**B**); scale bar 10 µm; quantification of perilpin levels of protein expression (**C**).

**Table 1 materials-12-03220-t001:** Preparation conditions and phosphate groups content of HP/poly(vinyl alcohol) (PVA) composite hydrogels.

Sample Code	Composition P/PVA (g %)	Crosslinking Method		Phosphate Groups Content * (mmol/g Hydrogel)
		Procedure	Method	
**HP/PVA_47_-R** (average Mw ~ 47,000 g/mol, crosslinked at room temp)	50/50	Room temperature	chemical	1.93 ± 0.15
**HP/PVA_47_-F** (average Mw ~ 47,000 g/mol, crosslinked by cryogelation)	50/50	Cryogelation	Chemical and physical	1.39 ± 0.24
**HP/PVA_125_-F** (average Mw ~ 125,000 g/mol, crosslinked by cryogelation)	75/25	Cryogelation	Chemical and physical	1.73 ± 0.26

* Determined by conductometric titration with 0.1N HCl.

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
