# Peer review of "Pullulan/Poly(Vinyl Alcohol) Composite Hydrogels for Adipose Tissue Engineering"

_materials, 2019, doi:10.3390/ma12193220_

Round 1

Reviewer 1 Report

In this manuscript, the authors synthesized pullulan/PVA composite hydrogels by both chemical crosslinking and chemical/physical dual crosslinking methods. They compared the swelling kinetics, dissolution behavior, and biocompatibility of the hydrogel scaffolds for tissue engineering. Overall, this manuscript is of good quality with well-designed experiments. And the publication of this paper will be beneficial for other researchers in this field. However, some minor problems need to be addressed before publication.

The words of this manuscript need to be checked carefully. In the abstract, we can see the words “extentthe”, “aresuitable”. In the method section, we can see some problematic sentence such as “The solution was cooled to room temperature and 1 mL of 10M NaOH and immediately after”. The abbreviations are not well explained in some cases (e.g. HP/PVA47-R, HP/PVA47-F in Table 1). And why there are both HP/PVA and P/PVA? The authors had better explain the naming to avoid confusion. In Figure 1, “A” and “B” are not marked in the pictures and scale bar is needed. In Figure 4B and Figure 5, the authors had better provide the sample number and statistical significance for the comparison. For Figure 6 and Figure 7, the authors had better provide the statistic methods in the method section.

Author Response

The words of this manuscript need to be checked carefully. In the abstract, we can see the words “extentthe”, “aresuitable”. In the method section, we can see some problematic sentence such as “The solution was cooled to room temperature and 1 mL of 10M NaOH and immediately after”.  

We corrected the sentences in manuscript.

The abbreviations are not well explained in some cases (e.g. HP/PVA47-R, HP/PVA47-F in Table 1).

Answer: For a better understanding we included a sentence at page 3, line 111 explaining the samples codification.

“The samples code is HP/PVAx-R or HP/PVAx-F, where P means pullulan, PVA is poly(vinyl alcohol), x= molecular mass of PVA in KDa, R and F refers to crosslinking procedure used (room temperature (R) or freeze-thawing (F)).”

And why there are both HP/PVA and P/PVA? The authors had better explain the naming to avoid confusion.   We corrected this error in the manuscript. Thank you for the observation.

In Figure 1, “A” and “B” are not marked in the pictures and scale bar is needed.  

Answer: This is right. Figure 1 was corrected.

In Figure 4B and Figure 5, the authors had better provide the sample number and statistical significance for the comparison.  

Answer: At page 3, line 126 it was already specified that “The experiment was performed in triplicate and average values were reported” for dissolution percent calculation. We included the sentence: “All data were averaged from three measurements” at page 4, line 142 for in vitro degradation studies.

A statiscal analysis was performed and the results are inserted in text at page 8, lines 273, 281 and 283.

For Figure 6 and Figure 7, the authors had better provide the statistic methods in the method section. 

The Authors thank you for this observation. We have now included the description of the statsistic methods used for biocompatibility assessment in the Materials and methods section of our manuscript, please find it enclosed at page 4:

2.2.8. Statistical analysis

                All experiments were performed in triplicate. Statistical analysis for biocompatibility and differentiation studies was performed using Graph Pad Prism software, one –way ANOVA method and Bonferroni correction. Statistical significance was considered for p<0.05.

Reviewer 2 Report

The authors present a work on developing composite hydrogels based on pullulan (P) and poly(vinyl alcohol) (PVA) for adipose tissue engineering. A few comments / queries to be addressed:

Abstract is concise and well-written Introduction provides sufficient background of the work What is the basis of the experimental design? From Table 1, three different groups are given - while the first two are HP/PVA-47-R and F respectively with P/PVA ratio of 50/50, the third one was HP/PVA-125 F with P/PVA ratio of 75/25. It would be more logical to have HP/PVA-125 F with 50/50 P/PVA ratio as one more group so that the P/PVA ratio is kept the same and the effect of PVA-47 and PVA-125 can be evaluated.  From Figure 4, it seems there is no significant variation in the dissolution percent between the three groups. Any reason? Or the significance bar is not added? Figure 6 - the scale bar text is not clear - add the scale bar value in the figure legend. Figure 6 - the live dead images, the morphology of the L929 cells seems to be rounded and not elongated? Figure 7 - why is the perilipin expression much higher in the control group than all the three experimental groups? There is no explanation in the text Conclusion can be improved with quantitative data

Author Response

What is the basis of the experimental design? From Table 1, three different groups are given - while the first two are HP/PVA-47-R and F respectively with P/PVA ratio of 50/50, the third one was HP/PVA-125 F with P/PVA ratio of 75/25. It would be more logical to have HP/PVA-125 F with 50/50 P/PVA ratio as one more group so that the P/PVA ratio is kept the same and the effect of PVA-47 and PVA-125 can be evaluated. 

Answer: The observation is right. Our aim was to compare the composite hydrogels properties induced by the synthesis procedure at the same composition and different molecular mass of PVA. Unfortunately, the hydrogel with P/PVA125 ratio of 50/50 was not possible to be obtained due to the high viscosity of PVA125.

From Figure 4, it seems there is no significant variation in the dissolution percent between the three groups. Any reason? Or the significance bar is not added?

 Answer: Three independent dissolution experiments were performed and the significance bar was already added in figure 4. The three groups of HP/PVA hydrogels are enough stable; however from the data presented in figure 4 it is obvious that by combining the two crosslinking methods (chemical and physical) a more stable structure is obtained. We performed a statistical analysis of our dissolution results and we found no significant difference between the three types of hydrogels. Consequently, the next sentence was introduced in text at page 8, line 273:

“However, the apparent difference between the dissolution percentages of the three types of hydrogels is not significant if SD value is considered.”

This fact can be due to the conditions in which the dissolution studies are performed. We believe that the double crosslinking procedure used for the hydrogels synthesis is responsible for their good stability.

Figure 6 - the scale bar text is not clear - add the scale bar value in the figure legend.

The Authors thank you for your observation. The scale bar is now more visible in figure 6 and the value was added to the figure legend.

Figure 6 - the live dead images, the morphology of the L929 cells seems to be rounded and not elongated?

The Authors thank you for your suggestion. The rounded morphology observed for the L929 cells cultured in contact with the scaffolds is due to the three-dimensional structure of the material. Cells tend to migrate and proliferate into scaffold’s pores, conserving a rounded phenotype due to the 3D environment. In contrast, you can observe a more elongated phenotype associated to the TCPS control, which is represented by a 2D environment that allowed cells to adhere and develop elongated cytoskeleton.

Figure 7 - why is the perilipin expression much higher in the control group than all the three experimental groups? There is no explanation in the text Conclusion can be improved with quantitative data 

The Authors thank you for your observation. The perilipin expression is much higher in the control group compared to the other three experimental groups because the control system used for this study was a tissue culture plastic system and the adipogenesis develops at a higher rate (higher speed and to a higher number of cells). We have now included the explanation in the manuscript and you can find it at page 11:

Perilipin expression was the highest on the control system because the cells seeded on tissue culture plastic system undergo adipogenic differentiation at a faster rate than the ones cultivated on materials.

Additionally, we have performed the quantification of TRITC-labeled perilipin from the confocal images (n=10 for each composition) and we have included a graph in figure 7 showing the levels of perilipin protein expression. The Authors thank you for this suggestion. We have also included a phrase in the manuscript in Results section:

When performing the quantification of the area and intensity of red-labeled perilipin by Image J, a clear higher expression of perilipin was obtained for HP/PVA125-F than on the other compositions, confirming that this scaffold supported adipogenesis with the highest efficiency.

Round 2

Reviewer 2 Report

All the comments are addressed.